# Chronic Partial Sleep Deprivation Increased the Incidence of Atrial Fibrillation by Promoting Pulmonary Vein and Atrial Arrhythmogenesis in a Rodent Model

**DOI:** 10.3390/ijms25147619

**Published:** 2024-07-11

**Authors:** Shuen-Hsin Liu, Fong-Jhih Lin, Yu-Hsun Kao, Pao-Huan Chen, Yung-Kuo Lin, Yen-Yu Lu, Yao-Chang Chen, Yi-Jen Chen

**Affiliations:** 1Graduate Institute of Clinical Medicine, College of Medicine, Taipei Medical University, Taipei 110, Taiwan; shuenhsinliu@gmail.com (S.-H.L.);; 2Division of Cardiology, Department of Internal Medicine, Shuang-Ho Hospital, Taipei Medical University, New Taipei City 235, Taiwan; 3Taipei Heart Institute, Taipei Medical University, Taipei 110, Taiwan; 4Department of Biomedical Engineering, National Defense Medical Center, Taipei 11490, Taiwan; 5Cardiovascular Research Center, Wan-Fang Hospital, Taipei Medical University, Taipei 116, Taiwan; 6Department of Psychiatry, Taipei Medical University Hospital, Taipei 110301, Taiwan; 7Department of Psychiatry, School of Medicine, College of Medicine, Taipei Medical University, Taipei 110, Taiwan; 8Division of Cardiology, Department of Internal Medicine, School of Medicine, College of Medicine, Taipei Medical University, Taipei 110, Taiwan; 9Division of Cardiovascular Medicine, Department of Internal Medicine, Wan-Fang Hospital, Taipei Medical University, Taipei 116, Taiwan; 10School of Medicine, College of Medicine, Fu Jen Catholic University, New Taipei City 24205, Taiwan; 11Division of Cardiology, Department of Internal Medicine, Sijhih Cathay General Hospital, New Taipei City 221, Taiwan

**Keywords:** sleep deprivation, heart failure, atrial fibrillation, G protein-coupled receptor kinase 2 (GRK2)

## Abstract

Sleep deprivation (SD) is a recognized risk factor for atrial fibrillation (AF), yet the precise molecular and electrophysiological mechanisms behind SD-induced AF are unclear. This study explores the electrical and structural changes that contribute to AF in chronic partial SD. We induced chronic partial SD in Wistar rats using a modified multiple-platform method. Echocardiography demonstrated impaired systolic and diastolic function in the left ventricle (LV) of the SD rats. The SD rats exhibited an elevated heart rate and a higher low-frequency to high-frequency ratio in a heart-rate variability analysis. Rapid transesophageal atrial pacing led to a higher incidence of AF and longer mean AF durations in the SD rats. Conventional microelectrode recordings showed accelerated pulmonary vein (PV) spontaneous activity in SD rats, along with a heightened occurrence of delayed after-depolarizations in the PV and left atrium (LA) induced by tachypacing and isoproterenol. A Western blot analysis showed reduced expression of G protein-coupled receptor kinase 2 (GRK2) in the LA of the SD rats. Chronic partial SD impairs LV function, promotes AF genesis, and increases PV and LA arrhythmogenesis, potentially attributed to sympathetic overactivity and reduced GRK2 expression. Targeting GRK2 signaling may offer promising therapeutic avenues for managing chronic partial SD-induced AF. Future investigations are mandatory to investigate the dose–response relationship between SD and AF genesis.

## 1. Introduction

The rising global prevalence of sleep disorders has garnered significant attention due to its adverse impact on human health. In addition to the widely recognized risk factor for atrial fibrillation (AF), which is sleep-disordered breathing, it has been demonstrated that a shorter sleep duration is also a risk factor for the incidence and prevalence of AF. Specifically, each one-hour reduction in sleep duration was associated with a 17% greater risk of prevalent AF and a 9% greater risk of incident AF [1] Human studies of total or partial sleep deprivation (SD) for a short period (less than 7 days) showed that SD activates the transcription of nuclear factor-κB in peripheral blood mononuclear cells and increases the level of circulatory proinflammatory cytokines, such as interleukin-6, tumor necrosis factor-α, interferon-γ, and C-reactive protein. Among healthy individuals, only one night of SD has been noted to prolong the QT interval and cause delays in both interatrial and intra-atrial electromechanical conduction. These observations imply that sleep disorders, aside from sleep-disordered breathing, could affect left atrial function and potentially play a role in the emergence of arrhythmias, including AF [2,3,4,5,6,7,8]. Moreover, chronic partial SD promoted sympathetic modulation and heart failure with increased cardiac fibrosis and hypertrophy in experimental animals [9]. However, the knowledge about the relationship between physiologic response during chronic SD and AF genesis is limited. Furthermore, there is a lack of direct evidence to establish a causal relationship between chronic partial SD and AF, including a comprehensive understanding of the molecular pathways and electrophysiological characteristics associated with chronic partial SD-induced AF triggers (PV) and substrate (LA). 

AF, the most common type of cardiac arrhythmia [10], is associated with increased risks of thromboembolism, heart failure, mortality after noncardiac surgery, poor quality of life, and mortality (overall 3.5-fold increase) [11,12,13]. AF occurrence is associated with atrial electrical and structural remodeling. Atrial electrical remodeling entails enhanced automaticity, premature depolarization, and re-entry, whereas atrial structural remodeling entails atrial enlargement, cellular hypertrophy, fibrosis, and myolysis with increased production of inflammatory markers [14,15]. Abnormal atrial calcium (Ca^2+^) handling has been proposed as a critical mechanism of atrial arrhythmogenesis. In a dog model, ventricular tachypacing-induced heart failure enhanced Ca^2+^/calmodulin-dependent protein kinase-II (CaMKII)-mediated phosphorylation of phospholamban (PLB), which further increased the sarcoplasmic reticulum (SR) Ca^2+^ load with spontaneous Ca^2+^ transients and triggered activity in the left atrium (LA). These events could be suppressed by inhibiting SR Ca^2+^ release or sodium (Na^+^)/Ca^2+^ exchanger (NCX) [16] On the other hand, the activation of CaMKII signaling with Ca^2+^ dysregulation in pulmonary vein (PV) cardiomyocytes promoted PV arrhythmogenesis. Moreover, arrhythmogenic PV cardiomyocytes may have larger late sodium currents (I_Na-Late_) and NCX currents [17,18,19]. These findings suggest a critical role of abnormal Ca^2+^ regulation in the pathophysiology of AF.

The putative crosstalk between the PVs and the sinoatrial node (SAN) has been proposed as a mechanism of atrial arrhythmogenesis. In an animal study, the PVs exhibited a higher number of burst firings and triggered activity in response to a provocative agent if the connection between the SAN and PVs was disrupted [20]. Furthermore, the loss of SAN overdrive suppression with subsequent increased PV burst firings in heart failure facilitates PV arrhythmogenesis [21]. In a canine model of pacing-induced heart failure, the suppression of the Ca^2+^ clock (spontaneous rhythmic release of Ca^2+^ from the SR) with attenuated intrinsic heart rate played a pivotal role in SAN dysfunction [22]. Nonetheless, it remains unresolved whether chronic partial SD might alter the overdrive activity of the SAN on the PV, potentially coupled with calcium mishandling, thus intensifying PV-related arrhythmogenesis and promoting the onset of AF. In this study, we utilized a well-established animal model of chronic partial SD to examine its impact on cardiac function and arrhythmogenesis in vivo. Additionally, we delved into the potential mechanisms behind the electrical and structural remodeling observed in chronic partial SD-induced AF triggers (PVs) and substrate (LA) through in vitro experiments.

## 2. Results

### 2.1. General Characteristics and Echocardiography and ECG Results

The SD and control groups had similar systolic BP (142.0 ± 7.5 vs. 139.7 ± 11.0 mmHg), diastolic BP (88.1 ± 8.3 mmHg vs. 92.8 ± 9.6 mmHg), and isolated heart weight (1.49 ± 0.04 g vs. 1.43 ± 0.05 g). However, the SD group had a significantly lower body weight (371.3 ± 8.4 g vs. 426.7 ± 5.1 g, *p* < 0.005) but a significantly higher heart weight-to-body weight ratio (4.0 ± 0.1 vs. 3.36 ± 0.16, *p* < 0.01) than the control group. Echocardiography revealed that the SD group had a larger LA diameter but a lower FS and stroke volume of LV than the control group (Figure 1). The SD group had a lower E/A ratio and tissue Doppler mitral e’ velocity with increasing E/e’ ratio than the control group, suggesting that the SD group had decreased systolic and diastolic functions of LV. In addition, the SD group had a lower TAPSE score than the control group, indicating decreased RV systolic function in the SD group.

The SD group had a significantly faster heart rate, longer QT, QTc, JT, T wave peak to T wave end (Tp-Te) intervals, and a larger T-wave amplitude than the control group (Figure 2A). The 24-h ambulatory telemetry ECG recording showed no spontaneous AF attack in the control group (0/6). However, one episode of sustained spontaneous AF was observed in one of the five rats in the SD group (Figure 2B).

A time-domain analysis of HRV revealed that the SD group had significantly lower SDNN and RMSSD than the control group. A frequency-domain analysis of HRV revealed lower HF power ms^2^, relative power of the HF band (HF power %), and HF power nu in the SD group than in the control group. In contrast, higher relative power of the LF band (LF power %) and higher LF power nu and LF/HF ratio were noted in the SD group than in the control group (Figure 3).

### 2.2. Transesophageal Electrophysiology Results

To estimate whether chronic partial SD predisposes rats to atrial arrhythmia, transesophageal atrial burst pacing was performed in both groups. The SD group had higher inducibility of sustained atrial arrhythmias (28% ± 8% vs. 6% ± 4%, *p* < 0.05) than the control group. The mean duration of sustained atrial arrhythmia was significantly longer in the SD group than in the control group (27.46 ± 8.44 s vs. 5.3 ± 3.25 s, *p* < 0.05; Figure 4).

### 2.3. Electrophysiological Characteristics of the PVs and SANs

As shown in Figure 5, five (83%) and four (67%) rats in the SD and control groups (6 rats per group) exhibited spontaneous PV activity. The PVs of the SD group had a higher beating rate than those of the control group (2.78 ± 0.24 Hz vs. 1.70 ± 0.32 Hz, *p* < 0.05). In contrast, the SANs of the SD group had a lower beating rate than those of the control group (3.13 ± 0.37 Hz vs. 4.12 ± 0.11 Hz, *p* < 0.05) (Figure 5A). Furthermore, the PVs of the SD group had higher incidences of DADs (80% vs. 0%, *p* < 0.05) and burst firing than those of the control group (Figure 5B).

### 2.4. Electropharmacological Characteristics of LA and RA

The LAs and the RAs of the control and SD groups had similar AP durations at repolarization extents of 20%, 50%, and 90% (APD_20_, APD_50_, and APD_90_, Figure 6A,B). As shown in Figure 6C, isoproterenol (1 μM) treatment and tachypacing (20 Hz) did not induce atrial tachyarrhythmia in the control LA but did induce it in the SD LA (0% vs. 66%, *p* = 0.06). The LAs of the SD group had a longer tachyarrhythmia duration than the control group. Furthermore, the LAs of the SD group exhibited a higher incidence of DAD than the control group (83% vs. 0%, respectively; *p* < 0.05). However, isoproterenol (1 μM) treatment and tachypacing (20 Hz) did not lead to any significant difference in the incidence of DAD (33.33% vs. 0%, *p* = 0.46) or atrial tachyarrhythmia (17% vs. 17%, *p* = 1.00) between SD RAs and control RAs.

### 2.5. Histopathology and Western Blotting Results

The SD group exhibited higher levels of fibrosis in the LA and the RA than the control group (Figure 7A). As shown in Figure 7B,C, SD LAs expressed a lower level of GRK2 than control LAs, but the expression levels of GRK2 were similar between control and SD RAs. Moreover, the expression level of β1-AR was similar between control and SD RAs and between control and SD LAs. We further evaluated the expression levels of Ca^2+^-handling proteins in the SD and control groups. We found that the levels of PKAc, Cav1.2 L-type Ca^2+^ channels, NCX, SERCA2a, PLB, p17-PLB, p16-PLB, RyR2, pRyR S2808, CaMKII, and pCaMKII did not vary significantly between control and SD RAs or between control and SD LAs. 

## 3. Discussion

Sleep is a vital physiological process that plays a crucial role in restoring the cardiovascular system, with increasing evidence suggesting that sleep disturbances may harm the cardiovascular system. However, contemporary lifestyles, work schedules, and alcohol consumption have gradually reduced the average sleep duration [23,24]. A population-based cohort study found that maintaining a healthy sleep pattern is linked to a reduced risk of AF [25].

In animal models of SD, both acute total and partial SD models have been frequently used to study cardiovascular effects. These models often employ modified single- or multiple-platform methods. The single-platform method involves placing an animal on a single small platform in an enclosed water tank, while the multiple-platform method involves placing grouped animals in a water tank containing more platforms than animals, with each platform being 6.5 cm in diameter. Previous literature evaluating the effects of different SD methods found no significant difference in paradoxical SD between the single- and multiple-platform methods [26]. Our study selected a modified multiple-platform method to induce chronic partial SD for a duration of 4 weeks to evaluate its impact on the molecular and electrophysiologic effects of AF genesis.

Sympathetic activation has been identified as a key factor linking sleep disturbances to cardiovascular disease. The effects of acute total SD on the autonomic nervous system have yielded conflicting results; some studies report a reduction in sympathetic activity, while others show an increase [27,28]. Animal studies have shown that acute total SD for 3–7 consecutive days, without interrupting the resting phase, leads to increased inducibility of AF via transesophageal atrial burst pacing [29]. Additionally, acute total SD is associated with increased cardiac fibrosis and hypertrophy, along with elevated levels of oxidative stress and inflammatory markers in the plasma [30,31]. Acute total SD is also associated with increased ventricular arrhythmogenesis and contributes to decreasing the current densities of Ito and the prolonged AP duration of the ventricular cardiomyocyte [32,33]. 

However, acute total SD is non-physiologic and corresponds less to a modern lifestyle. Thus, the impact of mild chronic partial SD that mimics real life on atrial arrhythmogenesis remains unclear. To the best of our knowledge, this study marks the first report of a spontaneous AF attack in a chronic partial SD model aimed at mimicking chronic sleep disturbance in the real world. It establishes a direct causal relationship between chronic partial SD and the initiation of AF, along with the molecular mechanisms and electrophysiological characteristics of AF triggers and substrates.

### 3.1. SD Induced Sympathetic Overactivity, Biventricular Dysfunction, and Increased Atrial Fibrosis

Human cardiac functions, including inotropic, lusitropic, dromotropic, and chronotropic effects, are tuned sophistically by the autonomic nervous system to meet physiologic needs and respond to a “fight-or-flight” response. Excessive sympathetic activity has been proven to be associated with cardiac injury. In animal studies investigating catecholamine-induced cardiotoxicity, chronic infusion of isoproterenol, a beta-adrenoreceptor agonist, led to increased myocardial apoptosis, left ventricular hypertrophy, dilation, and pump failure in rats [34]. Heart rate variability is currently one of the most promising quantitative markers of autonomic activity in both human and animal studies, with a high LF/HF ratio indicating sympathetic dominance [35,36]. In this study, we demonstrated that chronic partial SD could induce sympathetic overactivity, as evidenced by the increased LF/HF ratio and increased heart rate in the HRV and ECG studies, respectively. This sympathetic overactivity may contribute to LV hypertrophy, increased LV fibrosis, as well as LV dysfunction in both diastole and systole, as proved by the echocardiography.

Previous studies showed that LV dysfunction was linked to a substantial elevation in the risk of AF, and through mechanoelectrical feedback and neurohumoral modulation, LV diastolic and systolic dysfunction may result in the dilatation and increased fibrosis of the LA with increased LA vulnerability to AF [37,38]. Cardiac fibrosis, a detrimental factor linked to almost all types of heart disease, occurs when cardiac fibroblasts become overactivated, leading to the excessive buildup of extracellular matrix proteins. Consequently, this process reduces tissue compliance, disrupts electrical conduction, and accelerates the progression toward heart failure [39,40]. Previous studies have demonstrated that acute and chronic SD rats exhibit enhanced profibrotic signaling and experience cardiac abnormalities, including myocardial hypertrophy, interstitial fibrosis, and pumping failure [30,41,42]. In our recent study, SD rats exhibited impaired contractility and increased fibrosis in the LV, which may be caused by their higher expression levels of transforming growth factor beta (TGF-β) and phosphorylated Smad 2/3 [9]. In alignment with these results, our current study observed that chronic partial SD induced both LV systolic and diastolic dysfunction, as well as RV systolic dysfunction. Additionally, it increased the fibrotic load in the LA and RA, creating a substrate that facilitates the initiation and perpetuation of AF.

### 3.2. SD Induced Sinus Node Dysfunction and Increased Delayed Afterdepolarization (DAD) in AF Triggers and Substrate

PVs have been established as significant contributors to the initiation of AF. The mechanisms proposed for AF initiation in PVs include re-entry, triggered activity, and abnormal PV automaticity [43]. PV cardiomyocytes exhibiting spontaneous activity have a less negative resting membrane potential, influenced by perivascular adrenergic and cholinergic stimulation. The spontaneous PV activity is suppressed by faster SAN activity due to overdrive suppression [44]. Abnormal intracellular Ca^2+^ handling plays a vital role in generating spontaneous PV activity [45]. Emerging evidence suggests that SAN dysfunction exacerbates the occurrence of PV arrhythmogenesis, potentially playing a role in the heightened risk of AF observed in individuals with sick sinus syndrome. When the connection between the SAN and PVs was disrupted, the PVs demonstrated a greater occurrence of burst firings and triggered activity in response to provocative agents than the control SAN–PV preparation with an intact SAN–PV connection [20]. Additionally, heart failure substantially affects the electrical activity of the SAN [46].

Elevated intracellular Ca^2+^ concentration activates the forward-mode Na^+^-Ca^2+^ exchanger, which extrudes Ca^2+^ from the cytoplasm, generates an inward current, and slowly depolarizes the cell membrane to reach the threshold, leading to the genesis of triggered activity. β-adrenergic stimulation may increase cellular Ca^2+^ loading and the ryanodine receptor 2 open probability via PKA and CaMKII phosphorylation [47]. Our current study found that increased spontaneous PV activity and triggered PV activity in SD rats may be due to enhanced cellular Ca^2+^ loading caused by heightened adrenergic activity around PV cardiomyocytes. Furthermore, in comparison with isolated PV, severe heart failure resulted in decreased spontaneous activity in isolated SAN tissue at baseline and after isoproterenol infusion in SD rats. Under conditions of sympathetic overactivity in SD rats, this study suggests that the slower SAN activity cannot provide overdrive suppression to the anatomically normal PVs connected to the atrium. This lack of suppression may further augment the PV’s triggered activity and abnormal automaticity, serving as an initiator of AF in SD rats.

### 3.3. SD Results in Down-Regulation of G Protein-Coupled Receptor Kinases 2 (GRK2) Expression 

When exposed to prolonged sympathetic stimulation, cardiac β1-adrenoceptors (β1-ARs), which belong to the G-protein-coupled receptors (GPCRs) superfamily, undergo agonist-induced (homologous) desensitization and down-regulation. These processes are initiated by receptor phosphorylation, carried out by a family of serine/threonine kinases known as G protein-coupled receptor kinases (GRKs), with GRK2 being implicated in the gradual depletion of the cardiac inotropic reserves in chronic heart failure [48]. In the case of chronic heart failure, the high levels of catecholamines from the autonomic nervous system stimulating β1-ARs induce an increase in the expression of GRK2 within cardiomyocytes. Following the phosphorylation of β1-ARs by GRK2, β-arrestins bind to the phosphorylated receptor, and β-arrestins not only dissociate receptors from heterotrimeric G proteins (functional desensitization) but also mediate the internalization of numerous β1-ARs through clathrin-coated vesicles (downregulation) [49,50]. This enhanced GRK2-mediated cardiac β1-ARs down-regulation leads to approximately 62% “selective” down-regulation of the β1-ARs subpopulation in failing human hearts [51]. This regulatory mechanism reduces the receptor’s responsiveness to persistent or repeated agonist activation, ultimately depleting the heart’s inotropic reserve [52]. While this rise in GRK2 levels could function as a protective mechanism, aiming to safeguard the heart against potential cardiac toxicity induced by excessive catecholamines, an animal study demonstrated that cardiac-specific overexpression of GRK2, mirroring the upregulation levels seen in human heart failure (i.e., a 3-fold to 4-fold increase), significantly diminished β1-ARs’ signaling and cardiac contractile reserve [53]. It may explain at least in part why β-blockers currently, by decelerating β1-ARs desensitization and increasing β1-ARs density, the primary treatment for patients with chronic heart failure, leads to improving systolic function and reducing mortality and morbidity [54]. Contrary to expectations, animal studies also found that cardiac-specific GRK2 knockout mice exhibited accelerated progression of catecholamine toxicity when treated chronically with isoproterenol [55]. These findings showed that cardiac GRK2 is an absolutely critical regulator of cardiac β1-AR-dependent contractility and function. Maintaining an optimal level of GRK2 is imperative in various stress situations to ensure proper cardiac performance. Interestingly, our study found that despite chronic sympathetic overactivity and overt heart failure in SD rats, the SD LA did not exhibit compensatory overexpression of GRK2 along with β1-ARs downregulation. Instead, there was a significant decrease in GRK2 expression accompanied by normal β1-ARs density. Consequently, chronic and uncontrolled adrenergic stimulation in SD LA leads to an increased risk of atrial fibrosis, calcium dysregulation, and DADs and promotes the onset and persistence of AF in SD rats. Accordingly, our research exposed sympathetic overactivity in companies with down-regulated GRK2 expression may play critical roles in AF initiation and perpetuation in chronic partial SD-induced heart failure and AF. It may present the potential for introducing GRK2 expression modulation as an innovative treatment strategy to restore compromised atrial myocyte function and prevent AF attacks for patients with chronic sleep disturbances complicated by heart failure and AF. (Figure 8).

### 3.4. Study Limitation

Our study is subject to certain limitations. Our study primarily aimed to determine whether chronic partial SD is related to spontaneous attacks of AF. After a 4-week duration of chronic partial SD, we used a radiotelemetric transmitter surgically implanted subcutaneously in the abdomen to record ambulatory electrocardiography and detect spontaneous AF. To avoid surgical stress, risk, and experimental bias in the SD rats, other physiological recordings using additional surgically implanted telemetry devices, including polysomnography, were avoided during the SD period. Consequently, we cannot provide data on various other physiological parameters, such as heart rate, blood pressure, and body temperature of the SD rats during the SD period, nor can we accurately measure the decline in individual sleep stages to establish correlated treatment effects of the SD. The use of polysomnography to assess sleep loss stage-by-stage would have been a more suitable approach for elucidating the complex cause-and-effect relationship between SD and atrial arrhythmogenesis [56]. Furthermore, the modified multiple-platform method with limited and temporary SD might have allowed sufficient recovery, potentially reducing sympathetic hyperactivity during rest without inducing sustained sympathetic overactivity. This could explain the similar levels of expression and phosphorylation of calcium-handling proteins observed between the control and test groups. Additionally, without multiple test groups exposed to different levels of SD (e.g., varying durations or severities), it is difficult to determine whether the observed effects are specific to the exact conditions, and we could not establish a dose–response relationship between SD, GRK2 expression, atrial arrhythmogenesis, and cardiac function. It is also challenging to isolate the effects of SD from other potential stressors introduced by the experimental design, such as falling into the water or the fear and stress associated with the experimental setup. Nor can it determine a threshold effect of SD on AF genesis. Future investigations with multiple test groups incorporating varying durations of SD (including short-term, prolonged, partial, and total SD) and employing cardiac-specific GRK2 overexpression models in rats could offer a dose–response relationship between SD and AF genesis and provide a more precise understanding of the pathological roles of GRK2 and calcium-handling proteins in SD-induced atrial arrhythmogenesis. Moreover, our animal model may not fully replicate the effects of sleep disorders observed in clinical settings, emphasizing the importance of extending these findings to real-world proteomic analysis.

## 4. Methods

### 4.1. Animal Preparation, Housing and Care

The research study was conducted in adherence to the principles and guidelines outlined in the Declaration of Helsinki, which ensures the ethical conduct of medical research. Additionally, the study protocol was thoroughly reviewed and approved by the Institutional Animal Care and Use Committee of Taipei Medical University, Taiwan (approval number: LAC2022-0451). Male Wistar rats (age, 2 months; weight, approximately 250–350 g) were purchased and maintained at the Laboratory Animal Center of Taipei Medical University of Taiwan under the following standard conditions: 12-h light/dark cycle, constant temperature (+22 °C), housed in pairs to maintain their social stability and ad libitum food and water. Our animal facility staff performed cage cleaning following standard procedures. 

### 4.2. Chronic Partial SD Procedure

The rats used for the experiment were randomly divided into the SD group and the control group. Rats assigned to the control group were housed in standard environmental conditions as described previously and maintained on commercial rat chow and tap water ad libitum. The rats assigned to the SD group were subjected to chronic partial SD for 16 h per day (16:00–08:00) over 4 weeks, and during the remaining 8 h (8:00–16:00), the rats were placed back in their home cages. As described, the modified multiple-platform method was used to establish chronic partial SD at the Laboratory Animal Center of Taipei Medical University [9]. The SD rats were placed into a water tank (length, 100 cm; width, 45 cm; and height, 30 cm) containing 10 round platforms (diameter, 6 cm; height, 10 cm) positioned in 2 rows (Appendix A). The platforms were fixed at the bottom of the tank at a distance of 8 cm, and water was filled up to 2 cm below the platform height. The rats could move freely on the platforms to obtain food and water throughout the study period. When the rats fell asleep, they lost their muscle tone and fell from the platform into the water, which would wake them; thus, the rats were deprived of sleep. This method results in a 100% loss of paradoxical sleep and a 37% loss of slow-wave sleep [26,57]. The tank water was changed daily. Appendix A presents the experimental timeline for the two groups. The electropharmacology study of isolated tissue preparations was conducted at the National Defence Medical Center, and other animal experiments were conducted at Taipei Medical University of Taiwan. 

### 4.3. Non-Invasive Blood Pressure Measurement Using the Tail-Cuff Method

At the end of the four-week-long treatment, the blood pressure (BP) and heart rate of all rats (n = 6 in each group) were measured (time: 13:00–17:00) at the standard 25 °C room temperature using a non-invasive BP monitoring kit (MK-2000ST; Muromachi Kikai Co., Ltd., Tokyo, Japan) in accordance with the manufacturer’s instructions. To measure BP under relatively stress-free conditions, the rats were exposed to the measurement conditions for 15 min/day for 7–14 days before the final experiment. For this, a dark brown acryl holder was used to restrain the animal, and a cuff with a pneumatic pulse sensor was attached to its tail. BP and heart rate were photoplethysmographically measured by assessing a pulsatile tail artery, and the average of at least three consecutive readings obtained from each rat was used for analysis [58,59].

### 4.4. Surface Electrocardiography, Ambulatory Electrocardiography, and Power Spectral Analysis of Heart Rate Variability

After a four-week-long treatment, surface electrocardiography (ECG) was performed for all rats (age, 12 weeks; n = 6 in each group). Light anesthesia was induced through 5% isoflurane inhalation. After the animal was stabilized, silver needle electrodes were placed under the skin of the forelimbs and the left hind limb. Surface ECG signals were continuously recorded using PowerLab C (ADInstruments Pty Ltd., Bella Vista, NSW, Australia) for at least 2 min in the lead II configuration. LabChart (version 8; ADInstruments Pty Ltd.) was used for data analysis.

After a week, an ambulatory ECG was performed for conscious, freely moving rats (age, 13 weeks; n = 5 in the SD group; n = 6 in the control group). After deep anesthesia was induced through 5% isoflurane inhalation, a DSI radiotelemetric transmitter (model ETA-F10; Data Sciences International, St. Paul, MN, USA) was surgically implanted subcutaneously in the abdomen 3 days before the rats were euthanized. The negative and positive leads were placed and tacked (using a suture) to the right upper chest and left abdominal muscles, respectively, in a lead II configuration [60]. The rat was observed for a 48-h recovery period to prevent surgical complications; when the animal regained consciousness, it was placed on a small animal receiver (RPC-1; Data Sciences International, St. Paul, MN, USA) to record ECG signals in a dark room with attenuated background noise.

Short-term (5 min) ECG recordings were obtained to assess heart rate variability (HRV) after 1 h of adaptation. Overnight (23 h) ambulatory ECG data were collected to determine if any spontaneous AF attack occurred. The ECG data were analyzed using LabChart (version 8; Data Sciences International, St. Paul, MN, USA), and a power spectral HRV analysis was performed using Kubios HRV Standard (Kubios Oy, Kuopio, Finland). Time-domain and frequency-domain analyses of HRV were performed. The parameters estimated in the time-domain analysis included the standard deviation of normal to normal R–R intervals (SDNN) and the root mean square of successive R–R interval differences (RMSSD); the parameters estimated in the frequency-domain analysis included the absolute power of the low-frequency (LF) band (LF Power ms^2^), relative power of the LF band in normal units (LF Power nu), absolute power of the HF band (HF Power ms^2^), relative power of the HF band in normal units (HF Power nu), and total power (ms^2^). Two frequency bands were used: LF (0.20–0.75 Hz) and HF (0.75–2.50 Hz) bands [36,61].

### 4.5. Transthoracic Echocardiography

A transthoracic echocardiography was performed for rats aged 12 weeks (n = 6 in each group) using our previously described method [62]. For this, the rats under isoflurane anesthesia (5% for induction and 2% for maintenance) were examined using Vivid i Portable Cardiac Ultrasound (GE Healthcare, Haifa, Israel). M-mode tracing of the left ventricle (LV) and LA was performed to measure LV wall thickness at end diastole, LV internal diameter at end diastole and end systole, LV fractional shortening (FS), LV ejection fraction (EF), and LA diameter. LV diastolic function was evaluated using a pulsed-wave Doppler; the mitral inflow pattern (E and A waves) and the tissue Doppler velocity at the medial mitral annulus (e wave) were analyzed. Systolic dysfunction of the right ventricle (RV) was assessed using the tricuspid annular plane systolic excursion (TAPSE) scoring system.

### 4.6. Transesophageal Electrophysiology

After 4 weeks of SD treatment, a closed-chest transesophageal electrophysiology was performed (n = 6 in the SD group; n = 5 in the control group) using a previously described method [63]. Briefly, light anesthesia was induced via 5% isoflurane inhalation. A traditional surface ECG (lead II configuration) was performed using stainless steel needle electrodes (EL451; BIOPAC Systems Inc., Goleta, CA, USA) and analyzed with a data acquisition system (MP36; BIOPAC Systems Inc.). An octapolar catheter (1.1-French, FTS-1913A-1018; Transonic Scisense Inc., London, ON, Canada) was inserted through the oral cavity into the esophagus for atrial stimulation from a distal electrode pair. The correct position of the atrial stimulation catheter was confirmed by achieving constant atrial capture, indicated by each electrical stimulus eliciting a narrow QRS complex, with electrical stimulation at a frequency of 450 stimuli per minute and a duration of 1 ms per stimulus. The pacing threshold was typically between 10 and 20 V. After establishing reproducible atrial pacing at twice the pacing threshold with a stimulation duration of 1 ms, an AF inducibility test was performed. This involved 10 burst pacing sessions (pacing cycle length of 25 ms, total duration of 30 s) with a 5-min free interval between bursts. The AF was characterized by rapid and fragmented atrial electrograms with an irregular ventricular rhythm sustained for at least 3 s post-pacing. The AF inducibility (number of AF episodes post-stimulation/total number of stimulations), cumulative AF duration, and mean AF duration were analyzed.

### 4.7. Electropharmacology in Isolated Tissue Preparation

An electropharmacology study of isolated tissue preparations was performed following our previously described method [64,65]. After the rats were anesthetized and euthanized using an overdose of inhaled isoflurane (5% in oxygen; Panion & BF Biotech, Taipei, Taiwan) for 10 min, the hearts of the rats were excised through a midline thoracotomy. The adequacy of anesthesia dosage was verified by the absence of corneal reflexes and motor responses to pain stimuli. Tissue preparations from LA (n = 6 in each group), right atrium (RA) (n = 6 in each group), SANs (n = 4 in the SD group; n = 6 in the control group), and PVs (n = 5 in the SD group; n = 4 in the control group) were isolated. Electropharmacological measurements were performed within two hours after the completion of the separation procedures. Tissue strips were perfused at a constant rate of 3 mL/min with Tyrode’s solution with the following composition: NaCl (137 mM), KCl (4 mM), NaHCO_3_ (15 mM), NaH_2_PO_4_ (0.5 mM), MgCl_2_ (0.5 mM), CaCl_2_ (2.7 mM), and dextrose (11 mM) in a 97% O_2_–3% CO_2_ gas mixture. The temperature was maintained at 37 °C, and the preparation was allowed to equilibrate for 1 h before electrophysiological analysis. The transmembrane action potentials (APs) of the PV, LA, SAN, and RA were recorded using machine-pulled glass capillary microelectrodes filled with KCl (3 M). The tissue preparations were connected to a World Precision Instruments FD223 Dual Differential Electrometer (Artisan Technology Group, Champaign, IL, USA) under a tension of 150 mg. Electrical activity was displayed on an oscilloscope (4072; Gould, Jafferson, OH, USA) and recorded using a recorder (TA11, Gould). Electrical stimuli were applied using an S88 Dual Output Square Pulse Stimulator (Grass Instruments, Norfolk, MA, USA) through the Grass SIU5B Stimulus Isolation Unit. Spontaneous activity was defined as the constant occurrence of spontaneous APs in the absence of any electrical stimulus. Delayed afterdepolarization (DAD) was defined as an afterpotential that occurs after the completion of repolarization and carries the membrane potential to a level more than that recorded later in diastole. Burst firing was defined as an accelerated spontaneous potential (faster than the basal rate) with sudden onset and termination. The LA and RA tissue preparations were perfused with isoproterenol (1 μM) to evaluate burst firing during high-frequency burst pacing (20 Hz) for 1 s.

### 4.8. Histopathological Analysis

After anesthesia was induced through 5% isoflurane inhalation, the rats (age, 13 weeks) were euthanized; no corneal reflex and no motor response to pain stimuli were ensured before euthanizing the rats. Then, their body weight was measured. Next, their hearts were carefully excised and weighed (n = 6 in each group). The heart weight/body weight ratio was calculated. The heart tissues were fixed with 4% paraformaldehyde, embedded in paraffin, and serially sectioned at 5 μm intervals. Tissue sections of LA (n = 4 in each group) and RA (n = 5 in each group) appendages were stained using Masson’s trichrome stains. Masson’s trichrome, a three-color staining method, is used to distinguish collagen in tissue. In this process, collagen fibers are stained blue, while the cytoplasm is stained red. Following the staining procedure, the collagen volume fraction—defined as the ratio of the total collagen surface area to the surface area of each individual chamber—was calculated to evaluate fibrosis severity. This analysis was performed using HistoQuest Analysis Software (version 4.0; TissueGnostics, Vienna, Austria) according to our previously established protocol [66].

### 4.9. Western Blotting

The LA (n = 6 in each group) and RA (n = 6 in each group) tissues were homogenized and centrifuged in buffer systems. Equal amounts of total protein were separated through 5% or 8% sodium dodecyl sulfate–polyacrylamide gel electrophoresis; the protein bands were transferred onto equilibrated polyvinylidene difluoride membranes. All blots were probed with primary antibodies against a beta-1 adrenergic receptor (β1-AR, #PA1-049, Thermo Fisher Scientific, Waltham, MA, USA), G protein-coupled receptor kinase 2 (GRK2, #SC-562, Santa Cruz Biotechnology, Dallas, TX, USA), Ca^2+^/calmodulin-dependent kinase II-δ (CaMKII-δ, #GTX111401, GeneTex, Irvine, CA, USA), phosphorylated CaMKII at Thr 286 (pCaMKII, #ab32678, Abcam, Cambridge, UK), ryanodine receptor type 2 (RyR2, #MA3-916, Thermo Fisher Scientific, Waltham, MA, USA), phosphorylated RyR2 at Ser 2808 (pRyR S2808, #A010-30AP, Badrilla, Leeds, UK), catalytic subunit of protein kinase A (PKAc, #610981, BD Transduction Laboratories, San Jose, CA, USA), Na^+^/Ca^2+^ exchanger (NCX, #R3F1, Swant, Burgdorf, Switzerland), Cav1.2 L-type calcium channel (Cav1.2, #ACC-003, Alomone Lab, Jerusalem, Israel), and glyceraldehyde-3-phosphate dehydrogenase (GADPH, #M171-7, MBL, Nagoya, Japan). All targeted bands were normalized to GADPH to confirm equal protein loading.

### 4.10. Statistical Analysis

Data are presented as mean ± standard error of the mean values for normally distributed variables and as frequency and percentage for categorical variables. Between-group differences were analyzed using an unpaired *t*-test (for normal distribution), a Wilcoxon signed-rank test (for non-normal distribution), and a Chi-square test (for categorical variables). Statistical analyses were performed using SigmaPlot (version 12; Systat Software, Inc., San Jose, CA, USA). The level of statistical significance was set at *p* < 0.05.

## 5. Conclusions

SD increases AF susceptibility by modulating the autonomic nervous system toward sympathetic predominance, resulting in heart failure with SAN dysfunction, suppressed GRK2 expression, and enhanced PV and LA arrhythmogenesis.

## Figures and Tables

**Figure 1 ijms-25-07619-f001:**
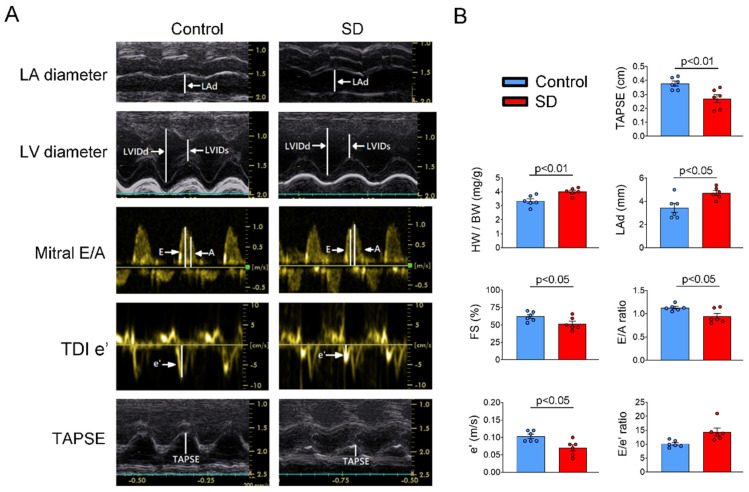
Echocardiographic study in the control and sleep deprivation (SD) rats. (**A**) Representative echocardiography images in the control and SD rats. (**B**) Average data of echocardiographic parameters in the control and SD groups. The SD group (n = 6) had a higher heart weight-to-body weight ratio and a larger left atrium (LA) diameter but lower left ventricular (LV) fractional shortening than the control group (n = 6). Compared with the findings in the control group, diastolic dysfunction was noted in the SD group with a reduced mitral inflow E/A ratio, reduced e’ of mitral annular tissue Doppler image, and increased E/e’ ratio. Finally, depressed right ventricular (RV) systolic function was observed in the SD group, as evidenced by a lower tricuspid annular plane systolic excursion (TAPSE) score than that of the control group. FS, left ventricular fractional shortening; HW/BW, heart weight-to-body weight ratio; LAd, left atrial diameter; LVIDd and LVIDs, left ventricular internal diameter end diastole and end systole; TAPSE, tricuspid annular plane systolic excursion; TDI, tissue Doppler image.

**Figure 2 ijms-25-07619-f002:**
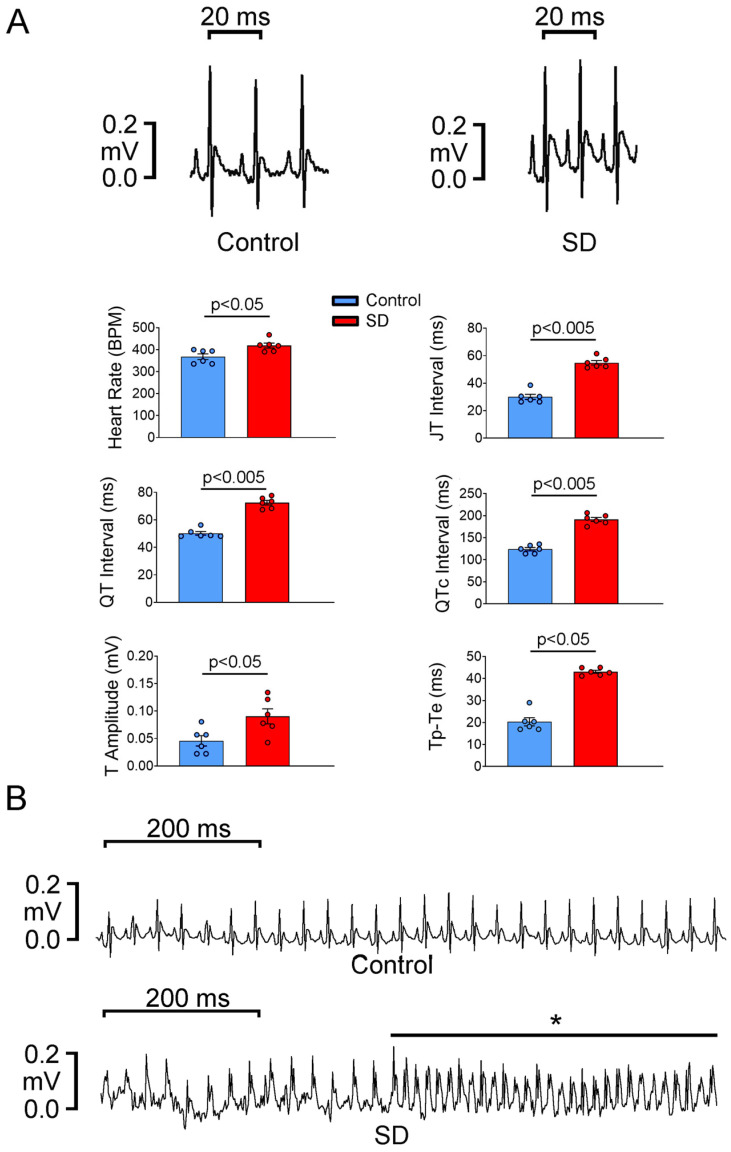
Results of surface electrocardiography (ECG; lead II) and 24-h ambulatory telemetry ECG performed for control and SD rats. (**A**) Surface ECG revealed an increased heart rate in the SD group than in the control group. The QT, QTc, JT, and T wave peak to T wave end (Tp-Te) intervals were significantly longer, and the T-wave amplitude was higher in SD rats than in control rats. (**B**) Overnight ambulatory telemetry ECG recording of the control and SD rats. A spontaneous attack of sustained AF (*) was noted in the SD group.

**Figure 3 ijms-25-07619-f003:**
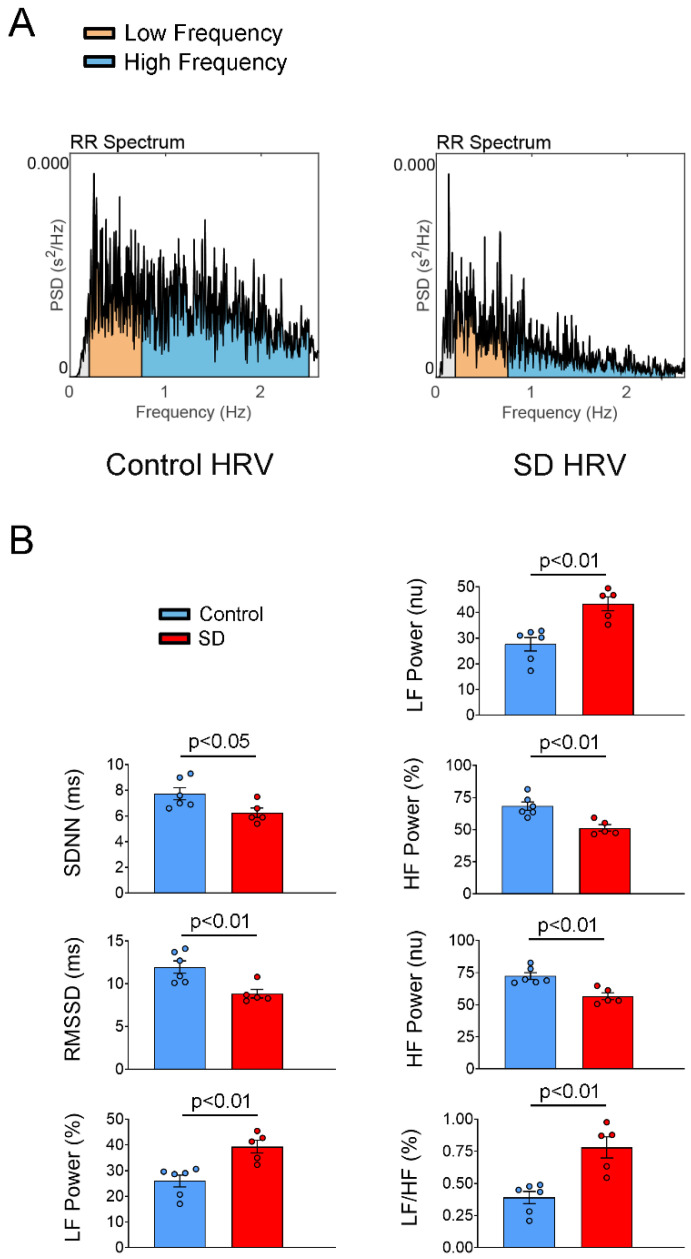
Heart rate variability (HRV) study in the control and SD groups. (**A**) Frequency-domain analysis of HRV was performed using Fourier transform-based Welch’s periodogram method; the results were in control (control HRV) and SD (SD HRV) rats. (**B**) Average data of time-domain and frequency-domain analysis of HRV. SD rats exhibited a lower standard deviation of normal to normal R–R intervals (SDNN), root mean square of successive R–R interval differences (RMSSD), relative power of the high-frequency (HF) band (HF Power %), and relative power of the HF band in normal units (HF Power nu) than did the control group. In contrast, the SD group exhibited higher relative power of the low-frequency (LF) band (LF Power %), relative power of the LF band in normal units (LF Power nu), and LF/HF ratio (LF/HF) than did the control group.

**Figure 4 ijms-25-07619-f004:**
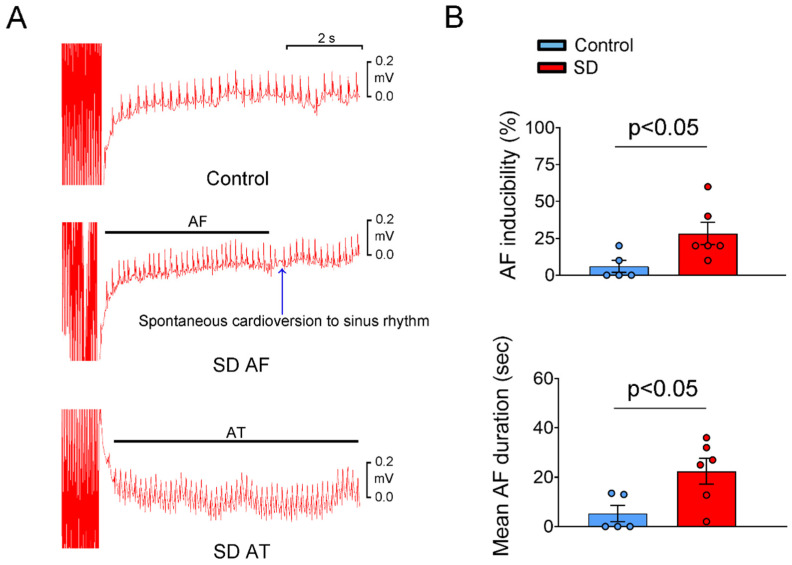
Results of transesophageal atrial pacing in the control and SD groups. (**A**) In the control group, transesophageal atrial pacing revealed that the sinus rhythm was maintained after rapid atrial pacing (pacing cycle length, 25 ms). In SD rats, prolonged AF with spontaneous cardioversion to sinus rhythm (SD AF) and sustained atrial tachycardia (SD AT) were observed after rapid atrial pacing. (**B**) SD rats exhibited significantly increased AF inducibility and longer mean AF duration compared to the control rats.

**Figure 5 ijms-25-07619-f005:**
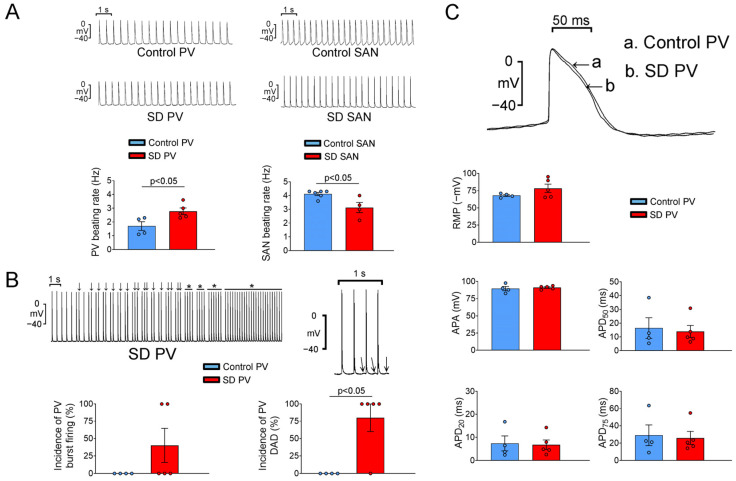
Pulmonary vein (PV) and sinoatrial node (SAN) electrical activities in the control and SD rats. (**A**) Recordings and average data of spontaneous activity in the PVs and SANs of the control and SD rats. SD PVs had a higher beating rate than control PVs. In contrast, SD SANs had a lower beating rate than control SANs. (**B**) Recordings showed delayed afterdepolarizations (DADs) and triggered activities (↓), as well as sustained burst firing (*) in SD PVs. SD PVs exhibited a significantly higher incidence of DADs and a higher incidence of burst firing than control PVs. (**C**) Tracing of action potential (AP) and the average data of AP parameters from control and SD PVs. RMP, resting membrane potential; APA, action potential amplitude; APD, action potential duration.

**Figure 6 ijms-25-07619-f006:**
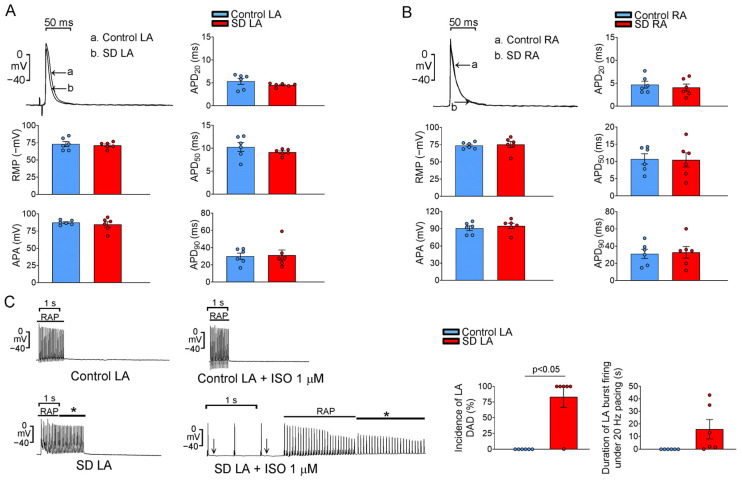
Electrophysiological characteristics and arrhythmogenesis of the left atrium (LA) and right atrium (RA) in control and SD rats. (**A**) Tracing of action potential (AP) and the average data of AP parameters from control (n = 6) and SD (n = 6) LAs. (**B**) Tracing of AP and the average data of AP parameters from control and SD RAs. (**C**) At baseline and after combined treatment with isoproterenol (1 μM) and tachypacing (20 Hz), delayed afterdepolarizations (DADs) (↓) and burst firing (*) could be induced in SD LAs but not in control LAs. The average data indicate that SD LAs exhibited a significantly higher incidence of DAD than control LAs. The duration of burst firing in SD LAs was longer than that in control LAs. RMP, resting membrane potential; APA, action potential amplitude; APD, action potential duration; RAP, rapid atrial pacing; and ISO, isoproterenol.

**Figure 7 ijms-25-07619-f007:**
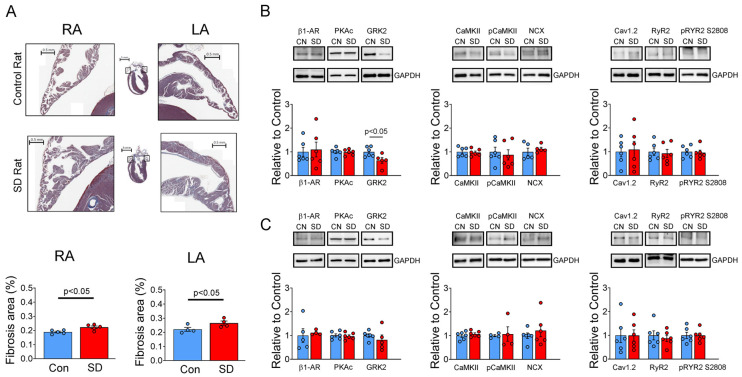
Histological and Western blot analyses were performed on the atrial appendages of both left atriums (LAs) and right atriums (RAs) in both control and SD rats. (**A**) Masson staining of the atrial appendages in the LAs and RAs of both the control and SD groups highlighted a significant increase in fibrosis and collagen deposition in the SD LAs and RAs when compared to the corresponding structures in the control group. (**B**) Expression and phosphorylation levels of calcium-handling proteins, upstream kinases, and β1-adrenoceptor (β1-AR) in the control and SD LAs. (**C**) Expression and phosphorylation levels of calcium-handling proteins, upstream kinases, and β1-AR in the control and SD RAs. The expression level of G protein-coupled receptor kinase 2 (GRK2) was significantly reduced in the LAs of SD rats in comparison to those of control rats. β1-AR, beta-1 adrenoceptor; CaMKII, Ca^2+^/calmodulin-dependent kinase II-δ; pCaMKII, phosphorylated CaMKII at Thr286; NCX, Na^+^/Ca^2+^ exchanger; Cav1.2, Cav1.2 L-type calcium channel; RyR2, ryanodine receptor type 2; pRyR S2808, phosphorylated RyR2 at Ser2808.

**Figure 8 ijms-25-07619-f008:**
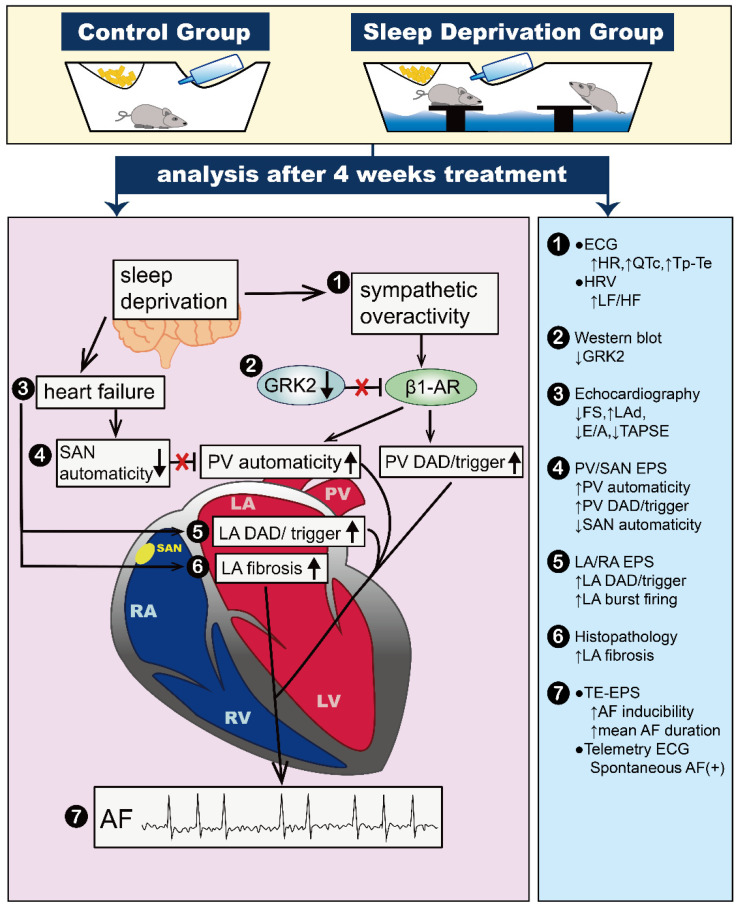
Schematic of the proposed molecular and electrophysiological mechanisms underlying chronic partial SD-induced AF: Chronic partial SD may lead to heart failure, contributing to left atrial (LA) enlargement, promoting LA fibrosis, and increasing the incidence of LA delayed afterdepolarizations (DADs) and triggered activities. Heart failure can also depress sinoatrial node (SAN) automaticity, leading to a loss of overdrive suppression in the pulmonary veins (PVs) of SD rats. On the other hand, chronic partial SD may induce sympathetic overactivity, accompanied by down-regulated expression of G protein-coupled receptor kinase 2 (GRK2), resulting in increased PV automaticity and a higher incidence of DADs and triggered activities in the PV. These mechanisms are likely to promote AF initiation and perpetuation in SD rats. Roman numerals represent pathophysiological changes and their corresponding experimental findings. β1-AR, beta-1 adrenoceptor; ECG, electrocardiography; E/A, mitral inflow E/A ratio; FS, left ventricular fractional shortening; HR, heart rate; HRV, heart rate variability; LAd, left atrial diameter; LF/HF, low-frequency to high-frequency ratio; QTc, QTc interval; TAPSE, tricuspid annular plane systolic excursion; TE-EPS, transesophageal electrophysiology.

## Data Availability

The data supporting this study’s findings are available from the corresponding author upon reasonable request.

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
