# Peer review of "Chronic Partial Sleep Deprivation Increased the Incidence of Atrial Fibrillation by Promoting Pulmonary Vein and Atrial Arrhythmogenesis in a Rodent Model"

_ijms, 2024, doi:10.3390/ijms25147619_

Round 1

Reviewer 1 Report

Comments and Suggestions for Authors

Congratulations to the authors for the work, which is characterized by a strong impact on the clinical landscape. However I have  some comments on your manuscript:Inizio modulo

Fine modulo

-            Why are the methods and statistical section located  after the results section?

-            Statistical section is absolutely not adequate for this kind of paper; please expand it properly;

-            Authors should  provide detailed descriptions of the control group's conditions to prove that they are comparable to the experimental group, apart from the SD intervention;

-            Authors should discuss the variability among animal models and how it was controlled or considered for  the analysis;

-             Add detailed protocols for electrophysiological recordings, including calibration steps, electrode placement verification, and measures taken to ensure consistency;

-            The paper should provide a more comprehensive description of the histological methods, including staining procedures, image analysis software used, and the criteria for quantifying fibrosis;

-            Please perform a power analysis to justify the sample sizes used, indeed proving  that the study is adequately powered to detect significant differences;

-            Correct the Inconsistent use of "SD" and "sleep deprivation."

Comments on the Quality of English Language

moderate english revisions are needed

Reviewer 2 Report

Comments and Suggestions for Authors

The study investigates how chronic partial sleep deprivation (SD) affects the risk of atrial fibrillation (AF) in rats. It used a platform method to induce chronic partial SD in Wistar rats and observed various cardiac functions. They found that SD led to impaired left ventricular (LV) function, increased heart rate, and higher susceptibility to AF. 

My major concern is that the paper only includes one test group, which underwent 16 hours of sleep deprivation per day, in addition to the control group. Without multiple test groups exposed to different levels of SD (e.g., varying durations or severities), it is difficult to determine whether the observed effects are specific to the exact conditions used in this study or if they would occur under other conditions as well. Additionally, without additional comparison groups, it’s challenging to isolate the effects of sleep deprivation from other potential stressors introduced by the experimental design, such as falling into the water or the fear and stress associated with the experimental setup.

Two further concerns regarding this are: first, the paper lacks a study on the dose-response relationship between different durations or severities of sleep deprivation and its effects on cardiac function. Identifying a dose-response relationship helps in understanding whether the risk of AF increases linearly with more severe or prolonged SD, or if there is a threshold effect. Second, the paper lacks a comprehensive sleep study that records various physiological parameters of the rats during sleep. These issues also impact the generalizability of the study's findings.

Round 2

Reviewer 1 Report

Comments and Suggestions for Authors

congratulations to the author for having improved so much their manuscript. You have answered to all revisions improving remarkably the overall quality of the paper.

Reviewer 2 Report

Comments and Suggestions for Authors

Thank you to the authors for their efforts in the revision. I believe the revision addresses my concerns by acknowledging the limitations. While it may not fully resolve all issues, it is acceptable given the challenges in conducting test groups under similar or identical conditions. I am open to accepting the paper if other reviewers are also in agreement.

Additionally, the explanation of the study's limitations, which highlights the current focus and the reasons behind it, should be synchronized in the abstract and introduction to provide readers with a better overview of the work.
